# Advances in Canaloplasty—Modified Techniques Yield Strong Pressure Reduction with Low Risk Profile

**DOI:** 10.3390/jcm12083031

**Published:** 2023-04-21

**Authors:** Peter Szurman

**Affiliations:** 1Eye Clinic Sulzbach, Knappschaft Hospital Saar, 66280 Sulzbach, Germany; peter.szurman@kksaar.de; Tel.: +49-06897-574-1119; 2Klaus Heimann Eye Research Institute (KHERI), 66280 Sulzbach, Germany

**Keywords:** canaloplasty, trabeculectomy, modifications, telemetric self-measurement, suprachoroidal drainage

## Abstract

For decades, trabeculectomy (TE) was considered the gold standard for surgical treatment of open-angle glaucoma owing to its powerful intraocular pressure (IOP)-lowering potency. However, owing to the invasive nature and high-risk profile of TE, this standard is changing, and minimally invasive procedures are becoming more preferable. In particular, canaloplasty (CP) has been established as a much gentler alternative in everyday life and is under development as a full-fledged replacement. This technique involves probing Schlemm’s canal with a microcatheter and inserting a pouch suture that places the trabecular meshwork under permanent tension. It aims to restore the natural outflow pathways of the aqueous humor and is independent of external wound healing. This physiological approach results in a significantly lower complication rate and allows considerably simplified perioperative management. There is now extensive evidence that canaloplasty achieves sufficient pressure reduction as well as a significant reduction in postoperative glaucoma medications. Unlike MIGS procedures, the indication is not only mild to moderate glaucoma; today, even advanced glaucoma benefits from the very low hypotony rate, which largely avoids a wipeout phenomenon. However, approximately half of patients are not completely medication-free after canaloplasty. As a consequence, a number of canaloplasty modifications have been developed with the goal of further enhancing the IOP-lowering effect while avoiding the risk of serious complications. By combining canaloplasty with the newly developed suprachoroidal drainage procedure, the individual improvements in trabecular facility and uveoscleral outflow facility appear to have an additive effect. Thus, for the first time, an IOP-lowering effect comparable to a successful trabeculectomy can be achieved. Other implant modifications also enhance the potential of canaloplasty or offer additional benefits such as the possibility of telemetric IOP self-measurement by the patient. This article reviews the modifications of canaloplasty, which has the potential to become a new gold standard in glaucoma surgery via stepwise refinement.

## 1. Introduction

Surgical treatment of glaucoma that cannot be controlled conservatively has undergone rapid transformation in recent years. Trabeculectomy (TE) has long been considered the gold standard globally; however, this surgical filtering technique, which dates back to the 1960s, is increasingly being questioned for several underlying reasons. First, the high risk of filtering bleb scarring can only be reduced with topical antimetabolites. Another disadvantage is the high rate of intra- and postoperative complications, ranging from bulbar hypotony with permanent visual fluctuations [1] and the risk of choroidal hemorrhaging [2], to an increased cataract rate [3] and the relevant lifelong risk of endophthalmitis [4]. These vision-threatening complications are largely due to the creation of an open connection between the anterior chamber and the subconjunctival space (penetrating surgery), which results in nonphysiologic and difficult-to-control bolus-like filtration instead of the desired continuous oozing effect. This is compounded by the time-consuming perioperative management and stressful follow-up care of the filtering bleb [5,6].

For this reason, there has always been a need for procedures with fewer complications and a comparable intraocular pressure (IOP)-lowering effect [7,8]. Although nonpenetrating procedures are, for the most part, significantly less complicating, none of the techniques investigated thus far appear to lower the IOP as effectively as TE. However, the introduction of canaloplasty (CP) and its gradual modifications have changed this picture. In the meantime, modified canaloplasty with additional suprachoroidal drainage (ScD) achieves IOP values similar to those of TE but without its complications.

## 2. Development and Rationale of Canaloplasty

Non-penetrating procedures are characterized by avoiding a direct connection between the anterior chamber and extraocular subconjunctival spaces in order to avoid the intractable problem of bleb scarring [9]. One of the first non-penetrating procedures was deep sclerectomy (DS), in which a continuous and controlled oozing of aqueous humor from the anterior chamber into an intrascleral pocket was achieved via the trabeculo-descemetic window (TDW) [10]. However, the further outflow remained unclear and was not sufficiently effective.

In parallel, Stegman developed viscocanalostomy (VC), which dilated Schlemm’s canal (SC) using a viscoelastic substance [11]. Thus, for the first time, a procedure existed that conceptually aimed to exclusively improve the natural drainage pathways (via SC and the adjacent collector channels).

Described in 2005 by Kearney, CP unites these two techniques: the DS approach allows a continuous ooze to be established using a prepared TDW, and VC is integrated to permanently—rather than temporarily—improve the physiological outflow by using microcatheter technology and a tension thread [12].

Thus, the rationale of CP is to activate and improve natural outflow pathways (trabecular facility) [12]. The IOP-lowering mechanism relies on several effects in the natural outflow tract (Figure 1):The SC is probed using a microcatheter (iTrack 250, Ellex Inc., Eden Prairie, MN, USA) for dilatation of the ostia and lumen of SC, and the adjacent collector channels using a viscoelastic. This effect dilates the canal to almost triple its original size, making it easily visible in the ultrasound biomicroscope (UBM) 50 MHz (Figure 1A).The placement of one or two 10.0 Prolene tensioning sutures results in permanent stretching and tightening of the trabecular meshwork (i.e., surgical “pilocarpine effect”) (Figure 1B) [12].Applying tension to the TDW results in the controlled percolation of aqueous humor into the intrascleral cleft formed after resection of the deep flap, which can be enhanced by YAG goniopuncture [13].

## 3. Surgical Technique of Conventional Canaloplasty

The detailed surgical technique has been described before [12]. In brief, the conjunctival opening is created superiorly at the limbus. In contrast to TE, cauterization should be avoided to preserve the episcleral vessels. First, a superficial scleral flap with 1/3 scleral thickness is dissected, with the incision extending to the clear cornea. The SC is unroofed after preparation of a second lamellar flap just above the choroidal plane, simultaneously creating a TDW. The deep scleral lamella is excised, forming an intrascleral lacuna and exposing the two ostia of the SC. After peeling off the juxtacanalicular TMW, the SC is probed and dilated using a 250 µm microcatheter under the constant administration of viscoelastic. The exact position of the catheter tip can be easily tracked via an illuminated laser light fiber located in the catheter, allowing the viscoelastic to be applied with accuracy. This viscodilation is an essential part of the procedure, as it breaks the adhesions within the SC, stretches the TMW, and perforates the intracanalicular septa, thus improving outflow into the collector channels.

After successful circular probing, a non-absorbable suture (Prolene 10.0) is fixed to the distal end of the microcatheter, inserted circularly by withdrawing the catheter into the SC, and knotted under tension. If available, a high-resolution UBM (Figure 1) can be performed at this stage to assess suture tension and dilation of the SC. The superficial flap is sutured to be watertight. Care is taken to avoid microfistulation under the conjunctiva or even the formation of a subconjunctival bleb (Figure 2) [12].

## 4. Spectrum of Severe Glaucomas Treated Using Canaloplasty

The indication for CP now includes all types of open angle glaucoma (OAG), including secondary OAG such as pigment dispersion, pseudoexfoliation, steroid-induced, uveitic, and juvenile glaucoma [14,15,16,17]. While earlier publications considered mild to moderate glaucoma as an indication for CP [18], it is increasingly being recognized that especially difficult, advanced, and preoperated glaucoma eyes are good indications for CP. These “difficult” glaucomas are characterized by high-grade pathologies with scarred conjunctiva and very high IOP values, in which all four natural outflow pathways are affected: they have a muddied TMW; a collapsed, fibrotic SC; narrowed collector vessels; and disrupted episcleral veins. At first glance, the high-grade pathology characteristic of severe glaucomas might argue against an intervention aimed at restoring the natural outflow pathway. Nevertheless, these eyes in particular often respond better to CP than to revisional filtering surgery [19,20].

There are several reasons for this: first, the higher the baseline pressure, the higher the IOP-lowering potency of CP [21]. In a study of 1000 canaloplasty patients, the IOP of eyes with a baseline IOP of >30 mmHg was reduced by more than 50% compared with eyes with a baseline IOP of <20 mmHg, which experienced only a 30% reduction [22].

Second, numerous studies have shown that CP can regulate secondary glaucoma as well as primary OAG. In particular, pseudoexfoliation glaucoma shows a very good outcome after CP, even in the long term [17].

Another argument is the very low hypotony rate of CP (<1%). This is particularly important for the often preterminal papillary situation of many difficult glaucomas and may help to prevent the wipeout phenomenon. On the other hand, TE has a very high hypotony rate of more than one-third, of which nearly 20% last longer than 3 months and carry a high risk of long-term damage to the optic nerve fibers [23] (Table 1).

An important new indication is preoperated eyes after failed TE [27,28,29]. At first glance, this indication seems unpromising because the SC integrity has been breached by TE, making probing difficult. In addition, the SC is known to collapse after filtering surgery and become dysfunctional from lack of use. Nevertheless, good evidence has shown that probing is usually possible even after TE, and the results are promising (Figure 3). In particular, eyes previously subjected to filtering surgery have highly altered conjunctiva, so they especially benefit from a bleb-independent approach.

## 5. Results of Conventional Canaloplasty

Several studies have shown that canaloplasty significantly lowers IOP over the long term with an excellent safety profile [18]. The average IOP reduction is 31–40%, reaching an average final level of 15.0–15.5 mmHg after 3 years [30,31,32]. There is a linear relationship between suture tension and IOP-lowering efficacy (Figure 4). Eyes with strong suture tension demonstrated in the UBM experienced a 50% greater IOP-lowering effect than eyes with low suture tension [33]. Interestingly, no difference was apparent between phakic and pseudophakic eyes, whereas eyes subjected to a combined phaco-CP (PCP) exhibited significantly lower IOPs with significantly less medication. The combined approach lowers the IOP by an additional 1.9 mmHg compared with using CP alone [34].

However, complete success (≤16 mmHg without medication) is achieved in only half the cases. This is because the number of topical glaucoma medications dropped significantly to 1.0–1.5, but only about half of all patients are medication-free after 3 years [14].

Comparative studies with TE confirmed that both methods achieved a sufficient IOP-lowering effect and significant reduction of postoperative glaucoma medications [7,8,23,35]. However, conventional CP (32–39% IOP reduction to 14.5 mmHg) did not reach the IOP level of successful TE (43–55% IOP reduction to 10.8 mmHg). A higher percentage of patients treated with CP rather than TE required postoperative medications (36% vs. 20%), but this did not reach significance [8]. Several meta-analyses confirmed a difference of only 2.3 mmHg between the two procedures at 12 months. However, when an IOP of ≤18 mmHg was considered as a criterion for success, no significant difference arose, neither for complete nor qualified success. On the other hand, the rate of visual-acuity-threatening complications was significantly increased in the TE group [20].

In summary, the consensus is that TE leads to a slightly greater reduction of IOP with a higher chance of being medication-free but has a higher risk profile, whereas CP patients required slightly more medication postoperatively but rarely had relevant complications.

These results can be interpreted in two different ways: on the one hand, the studies produced clear evidence that standard CP does not quite achieve the effect of TE in terms of absolute IOP reduction, but on the other hand, it is questionable whether such an extreme IOP reduction of up to 55% is really necessary or useful to these pre-damaged eyes [36]. The often-cited Wurzburg TVC study illustrates this dilemma very well. The authors found a relevantly higher risk profile at 43% IOP reduction after TE, which was accompanied by vision-threatening complications. The risk profile was lower at “only” 32% IOP reduction after CP [23]. In particular, eyes with several months of hypotony showed a high risk for a wipeout effect with permanent visual deterioration. Indeed, in the TE group, 37.5% experienced transient hypotony of <5 mmHg, and 18.8% had hypotony lasting longer than 90 days. Other hypotony-related complications of TE were choroidal detachment (12.5%) and shallow anterior chamber (6.2%). In addition, there were antimetabolite-associated complications such as corneal erosions and avascular filtering blebs [23].

Incidentally, if one excludes these hypotonic eyes in the TVC study, which cannot really be considered a success, the final IOP in the TE group is higher (no longer at 10.8, but at 12.8 mmHg) and does not achieve a better IOP reduction (–9.4 mmHg) than the CP group (–9.3 mmHg) [35]. Since the European Glaucoma Society (EGS) recommends only a 25% IOP reduction in the initial procedure, with an additional 20% reduction in the event of progression, merely aiming to maximize the IOP-lowering potential is too shortsighted [36,37].

This conclusion is also supported by a quality-of-life study that included 327 patients. The study showed higher patient satisfaction after CP compared to TE, which was statistically significant. This was due to better vision quality, fewer second procedures, significantly less stressful follow-up, and less impairment of quality of life [38].

Thus, this study mainly found that CP is not inferior to TE in terms of absolute IOP lowering, but that TE does offer a higher chance of achieving medication-free IOP lowering of <18 mmHg; however, this perk comes at the price of a higher complication rate with longer-term hypotony as well as a more complex follow-up [35].

## 6. Rationale for Modification of Canaloplasty

Despite the justification given above, it is undisputed that CP would benefit from a somewhat greater and more reliable IOP reduction. If CP could be modified to achieve a mean final IOP of, for example, 13 mmHg instead of 15 mmHg (i.e., within the range of successful TE), CP would likely replace TE as the gold standard owing to its significantly lower risk profile, easier follow-up, and lower impact on patient quality of life [7,23,38].

The goal of numerous scientific efforts, cited below, has been to modify CP in a manner such that it achieves IOP values comparable to those obtained after TE. In fact, numerous approaches have been devised to enhance the IOP-lowering potential of canaloplasty, which are described below.

## 7. Canaloplasty Combined with Suprachoroidal Drainage

One of the most promising modifications is to combine canaloplasty with an additional suprachoroidal drainage outflow (CP + ScD) to achieve an additive IOP-lowering effect by improving uveoscleral facility [39].

Previous techniques for non-penetrating glaucoma surgery, including conventional CP, have focused only on improving trabecular outflow. However, in addition to conventional trabecular outflow of aqueous humor [40], uveoscleral outflow also plays a significant role and may account for up to 57% of the total aqueous humor outflow under physiologic conditions [41]. It is important to exploit this potential.

Since conventional CP neglects the uveoscleral outflow pathway, the ScD technique was introduced in 2012 [39]. With this Sulzbach modification, the IOP-lowering effect of CP could be significantly enhanced and, for the first time, an IOP reduction comparable to TE could be achieved.

At first glance, the approach is similar to that of conventional CP. The crucial difference is that the deep scleral flap is not dissected lamellarly; instead, it is penetrated down to the choroid. This exposes the ciliary body and completely opens the suprachoroidal space. The deep flap is then dissected toward the limbus until the scleral spur is reached. The SC is safely unroofed and opened through a horizontal incision directly adjacent to the scleral spur (Figure 5). In the following the SC is probed analogously to conventional CP [39].

What is the purpose of the additional exposure of the choroidal window? First, the SC is much easier to find because blunt detachment to the scleral spur is simple, and the SC and TDW located directly anterior to the scleral spur can be prepared more reliably (Figure 5A,B). Second, percolation from the SC and TDW occurs not only into the collector vessels (trabecular outflow) but directly into the suprachoroidal space (uveoscleral outflow) via the choroidal window. The choroidal perfusion acts like a powerful water-jet pump and can reabsorb suprachoroidal fluid within minutes. The uveoscleral drainage effect is additive to the trabecular effect of conventional CP.

In a pilot study of 78 eyes, IOP was reduced to a mean of 13.5 mmHg after CP with ScD, with a concomitant reduction of 2.0 medications to 1.0 medication at 12 months. In addition, 52.6% of patients were completely medication-free [39].

These results were confirmed in a comparative, retrospective, two-arm study of 417 eyes over 12 months [24]. The mean IOP reduction after CP with ScD was significantly greater at 35.9% (from 20.9 ± 3.5 mmHg to 13.1 ± 2.5 mmHg) than after conventional canaloplasty at 31.2% (from 20.8 ± 3.6 mmHg to 14.0 ± 2.6 mmHg). The number of medications required was also lower (0.7 ± 1.0). The percentage of patients free of medication after one year was significantly higher in the combined group (56.9%) than in the conventional CP group (45.4%) [24].

Thus, the combination of CP with ScD lowered the IOP significantly more than conventional CP. More importantly, for the first time, the long-term IOP values are close to the IOP-lowering potential of TE, but free of serious complications.

## 8. Canaloplasty with ScD and Suprachoroidal Collagen Implant

To keep the outflow into the suprachoroidal space open in the long term, the additional implantation of a suprachoroidal collagen implant into the choroidal window was proposed in 2016 [42]. To achieve this, after the usual dissection down to the choroid at the end of surgery, the suprachoroidal space was widened with a viscoelastic, and a 10 × 10 × 2 mm collagen sponge (Ologen, Dahlhausen, Cologne, Germany) was implanted (Figure 6).

In a prospective study of 65 eyes over 12 months, the IOP-lowering effect was 35.6% (from 21.0 ± 4.3 mmHg to 13.5 ± 3.0 mmHg); the number of medications decreased from 3.5 to 0.9 [42]. In a recent study of 1034 eyes, this effect also seemed to be stable in the long term; after 2 years, the IOP was even slightly lower (12.9 ± 1.9 mmHg), suggesting a sustained effect [own data, unpublished]. This is remarkable because all large studies on conventional canaloplasty have shown a diminishing effect after 2 or 3 years.

Interestingly, modified CP with ScD and suprachoroidal Ologen also seems to be effective in treating pseudoexfoliation glaucoma. A retrospective study of 111 patients showed a stable IOP reduction of 45.8% to 12.7 ± 2.2 mmHg, even after 4 years. Again, there was no diminishing effect over time after Ologen implantation. This is in marked contrast to conventional CP, which has been judged in several studies to be less suitable for secondary glaucoma. Presumably, especially for secondary glaucoma, a purely trabecular facility improvement (standard CP) is insufficient, and the additional effect of uveoscleral outflow (ScD) is particularly useful [17].

In summary, suprachoroidal implantation of Ologen does not further enhance the IOP-lowering effect of CP with ScD, but the effect of ScD through a space holder in the choroidal window seems to be more sustainable and could maintain the achieved IOP level for at least 4 years.

## 9. Phacocanaloplasty with ScD

It has been well-acknowledged that combining CP with cataract surgery has an additive IOP-lowering effect. Comparative studies have shown that phacocanaloplasty (PCP) lowers the final pressure by an average of 1.3–1.9 mmHg more than CP alone [30,31]. However, this effect was less pronounced in other studies [32].

This additive effect has also been confirmed in studies employing CP with ScD. In a retrospective comparative study of 328 eyes, CP with ScD alone achieved a 37.0% reduction in IOP at 1 year (from 20.9 ± 3.6 mmHg to 13.2 ± 2.6 mmHg), whereas PCP with ScD produced a significantly greater reduction of 47.4% (from 23.2 ± 5.1 mmHg to 12.2 ± 1.7 mmHg) [43].

This “phaco” effect is independent of axial length and anterior chamber depth, unlike the IOP-lowering effect of cataract surgery in general. Only the preoperative IOP level has been shown to be a strong predictive factor: the higher the preoperative IOP, the greater the postoperative IOP reduction [43].

## 10. “Filtering” Canaloplasty with Bleb Formation

In CP, the scleral flap should be sutured to be as watertight as possible in order to avoid subconjunctival drainage with formation of a filtering bleb. Instead, drainage via Schlemm’s canal alone (and via ScD) is desired. This action makes CP immune to the scarring stimulus of the conjunctiva [44].

However, in clinical practice, a filtering bleb is sometimes formed after canaloplasty. Although it is hardly visible clinically, it can be detected by AC-OCT and UBM [45]. This discrete filtration zone often has no effect on postoperative IOP values but tends to be a negative prognostic factor.

On the other hand, one might assume that by creating an alternative outflow pathway for aqueous humor, an additional decrease in IOP can be achieved. To provide an additional subconjunctival outflow, the superficial scleral flap must be adapted in a dosed manner, similar to TE, to ensure continuous drainage of aqueous humor under the conjunctiva [46].

Technically, this “filtering” CP is not a non-penetrating surgical technique but rather a double-covered TE, which reduces the rate of hypotony but has all the disadvantages of a filtering surgery and so must be managed accordingly [46]. Therefore, to ensure the sustainability of a controlled filtration effect, the use of antimetabolites such as mitomycin C (MMC) as a sponge is reasonable (in analogy to TE). In his study population, Barnebey achieved an average IOP reduction of 42.7% and a reduction in medication from an average of 2.2 drugs to 0 at 12 months. The administered concentrations of MMC (0.025% and 0.03%) did not induce avascular areas, but the rate of postoperative hypotony (15%) was significantly higher than in the control group (1.1%) [46].

MMC can be applied on the superficial sclera, under the scleral flap, or in combination [47,48]. A meta-analysis showed better IOP reduction with MMC (43.56% at 6 months and 42.26% at 36 months) compared with surgery without MMC (39.14% at 6 months and 27.59% at 36 months). Complication rates for wound leakage, hypotony, expulsive hemorrhage, flattening of the anterior chamber, and cataract induction showed no significant differences. The authors conclude that CP with adjunctive use of MMC appears to be a way to improve the efficacy of standard CP [49].

## 11. Canaloplasty with ScD and Suprachoroidal Eyemate-SC IOP Sensor

In the future, an important modification is the combination of CP with a suprachoroidal IOP sensor for telemetric self-measurement by the patient. Particularly after glaucoma surgery, frequent and reliable measurement of the IOP is crucial to verify successful IOP adjustment in the target range [50,51].

The patented Eyemate-SC (Implandata, Hannover, Germany) is the first available suprachoroidal sensor for telemetric IOP self-measurement. It is placed in the suprachoroidal space during non-penetrating glaucoma surgery, where it remains permanently [52]. The suprachoroidal approach offers several advantages: first, the spatial separation of the IOP measurement from the site of pathology (chamber angle) prevents the causative glaucoma from worsening owing to the diagnostic implantation; second, the procedure can be performed regardless of the lens status and existence of any anterior chamber pathologies; and finally, implantation of the suprachoroidal device can be combined well with glaucoma surgery to monitor the therapeutic success and acts as a placeholder for ScD [50].

Suprachoroidal implantation of the Eyemate-SC IOP sensor (7.5 × 3.5 mm and an outwardly decreasing thickness of 1.3 mm in the center and 0.9 mm on the periphery) can be excellently combined with CP (Figure 7). Both procedures can be consecutively performed using the same access point. This is easiest in CP with ScD, since a choroidal window is prepared anyway, and the sensor can be used as a placeholder in the same way as the Ologen implant (https://youtu.be/F_p9iIGxB9U, accessed on 8 December 2018) [52].

Thus, the Eyemate-SC sensor not only serves as a placeholder in the suprachoroidal space but also allows continuous, postoperative IOP-monitoring simultaneously (Figure 8). This is particularly useful for glaucoma surgery patients. Patients undergoing glaucoma surgery usually have advanced visual field defects that require strict control of the mean IOP and diurnal IOP fluctuations. Although the latter have been shown to decrease using this approach, they still persist even after successful glaucoma surgery. Particularly in the postoperative phase (1–3 months), when the IOP fluctuates the most, the validity of Goldmann applanation tonometry (GAT) is limited owing to the altered corneal biomechanics [50,51].

The Eyemate-SC IOP sensor delivers a wireless readout of continuous IOP values. The Mesograph handheld device can store up to 3000 IOP readings. The data can be wirelessly transferred to a secure, web-based platform accessible to the supervising ophthalmologist. Since each measurement is stored with a time stamp, patients can automatically create their individual IOP profile to disclose short- and long-term fluctuations and, if necessary, the therapy can be adapted accordingly [53]. Thus, supervising ophthalmologists can base their therapy decisions on a broad database of many hundreds of measurements instead of only a few measurements per year.

Telemetric self-measurement using an intraocular sensor also has other advantages. The measured values correspond to the true IOP independently of the corneal biomechanics, and the active involvement of the patients as well as the direct treatment response are suitable to improve the poor therapy adherence in glaucoma patients [53,54].

The recently completed European multicenter pivotal study demonstrated excellent safety and reliably reproducible IOP measurements. Except for the early postoperative phase, the GAT and telemetric Eyemate-SC measurements were in excellent agreement, with a mean difference of 0.23 mmHg across all study eyes at 12 months [55].

## 12. Disadvantages of Modified Canaloplasty

Although one might assume that exposure of the suprachoroidal space could increase the risk of hemorrhage or hypotony, this was not confirmed in the studies. It could be shown that modified canaloplasty (with ScD and Ologen) has a comparably low risk profile as conventional CP. Vision-threatening complications in particular were absent, which seems advantageous compared to TE [24,42,43,55]. Only cutting the perforating vessels can cause intraoperative oozing bleeding temporarily obscuring the view, because cauterization of the choroid should be avoided.

It should be noted, however, that the combination of different outflow pathways makes it difficult to assess the contribution of the individual components. Furthermore, the different outflow pathways influence each other, which becomes most evident in “filtering” CP: When aqueous humor drains outwards, the influence of the other changes (trans-trabecular and suprachoroidal) decreases. Filtering CP with MMC also shows, analogously to TE, a significantly increased rate of postoperative hypotony [46].

Costs also differ significantly, as the microcatheter incurs significant additional costs. However, this is counterbalanced by the significantly lower preoperative effort, the lack of costs for MMC, a shorter hospitalization due to the absence of hypotony, and a simpler follow-up without extensive bleb care or bleb revisions (needling, 5-FU injections).

However, the main limitation is the lower scientific evidence (250 vs. 8000 references in PubMed). Most importantly, there are only a few comparative studies and no randomized clinical trials comparing modified CP with TE, so a final weighing of benefits and risks must await.

## 13. Conclusions

CP is a safe and effective glaucoma surgical procedure that—through various modifications—now achieves an IOP-lowering effect comparable to that associated with successful TE, but without the typical risk profile of TE. The combination of canaloplasty with the newly developed ScD enhances the IOP-lowering effect by adding a new uveoscleral drainage pathway under the choroid. By creating a choroidal window, percolation from the SC and TDW occurs not only into the collector vessels (trabecular outflow) but also directly into the suprachoroidal space (uveoscleral outflow). The choroidal perfusion acts like a powerful water-jet pump and can reabsorb suprachoroidal fluid within a few minutes. Therefore, the combination of CP with ScD lowers the IOP to a significantly greater extent than conventional CP. For the first time, it has been shown that the IOP reduction achieved by CP with ScD is sufficient for the treatment of severe glaucoma, allowing this combined approach to be used a primary procedure instead of TE (Figure 9). This shows that modified CP (+scD+Ologen) is on par with TE in terms of indication spectrum and IOP-lowering potency, clearly differentiating it from the newly emerged MIGS procedures, which show a lower effect on IOP and are reserved for mild to moderate glaucoma. Implant-based modifications implemented using a placeholder in the suprachoroidal space do not further enhance the IOP-lowering effect, but the effect of ScD seems to be more sustainable and could maintain treatment success for at least 4 years. In addition, the novel suprachoroidal IOP sensor allows safe telemetric IOP monitoring after successful canaloplasty.

## Figures and Tables

**Figure 1 jcm-12-03031-f001:**
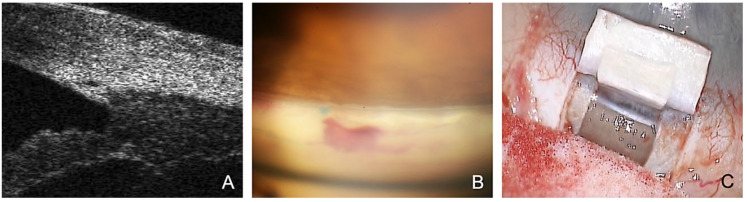
Rationale and mechanisms of action of canaloplasty. (**A**) Viscodilatation of the SC with microcatheter shown in UBM. (**B**) Prolene tension sutures are visible postoperatively on gonioscopy, resulting in permanent stretching of the TMW. (**C**) Controlled percolation of aqueous humor through the TDW.

**Figure 2 jcm-12-03031-f002:**
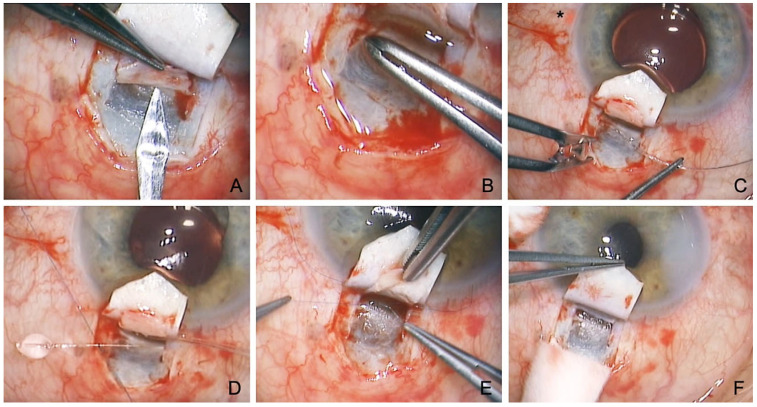
Surgical technique of conventional canaloplasty. (**A**) Unroofing of the SC after the incision of a second deep flap into the TDW. (**B**) Peeling of the juxtacanalicular TMW. (**C**) Probing of the SC with a microcatheter (Asterisk marks illuminated tip). (**D**) Insertion of two Prolene 10.0 sutures. (**E**) Knotting of the tension sutures. (**F**) Controlled percolation of aqueous humor through the TDW into the intrascleral cleft.

**Figure 3 jcm-12-03031-f003:**
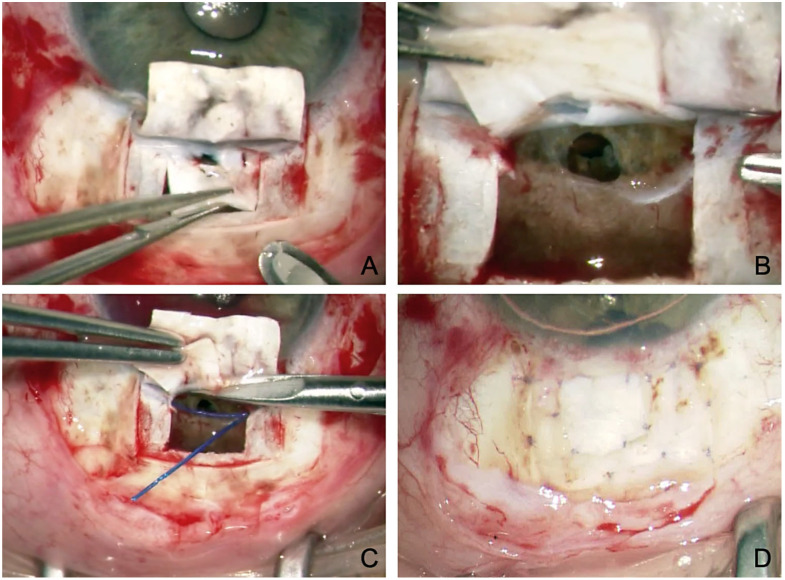
Revision canaloplasty after failed TE. (**A**) Incision and resection of the second deep flap. Note the atrophy in the superficial flap and the defect in the deep flap caused by the previous TE. (**B**) Visualization of the previous TE and iridectomy. (**C**) Probing of the SC on both sides using the suture canaloplasty technique. (**D**) Covering the superficial scleral atrophy with the resected deep scleral flap.

**Figure 4 jcm-12-03031-f004:**
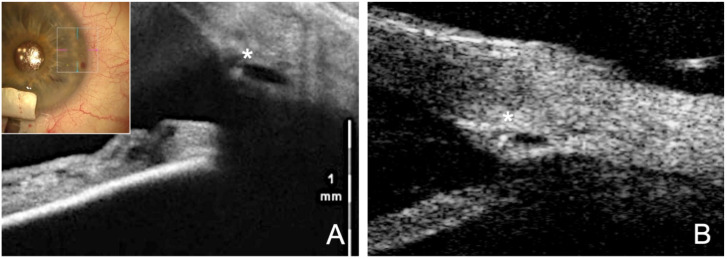
Visible effect of the tension suture stretching the SC. (**A**) Intraoperative visualization of the SC with intraoperative OCT. Asterisk marks the catheter tip within the lumen. (**B**) Postoperative UBM (50 MHz) clearly shows the dilated and stretched SC with knotted suture in the lumen (asterisk).

**Figure 5 jcm-12-03031-f005:**
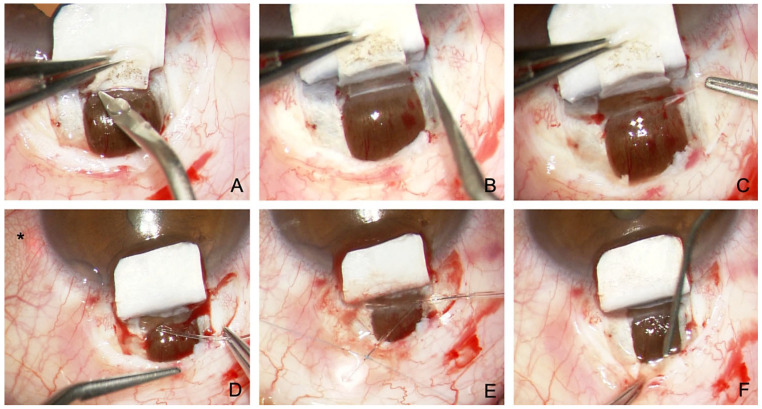
Surgical technique of modified canaloplasty with suprachoroidal drainage. (**A**) Preparation of a second deep flap with incision down to the choroid and incision anterior to the scleral spur. (**B**) Dissection across the scleral spur opens the SC and TDW. (**C**) Peeling of the juxtacanalicular TMW. (**D**) Probing of the SC with a microcatheter. Asterisk marks the laser light of the catheter tip (**E**) Insertion and knotting of two Prolene 10.0 sutures. (**F**) Dilatation of the suprachoroidal space using viscoelastics.

**Figure 6 jcm-12-03031-f006:**
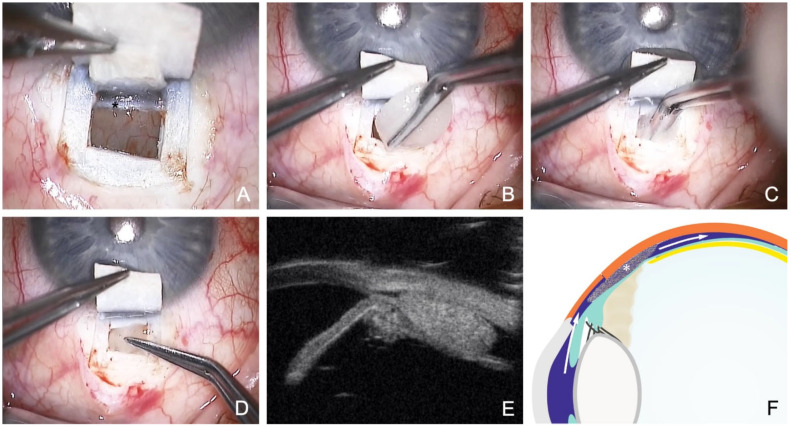
Canaloplasty with ScD and suprachoroidal Ologen sponge. (**A**) Clearly visible choroidal window, the scleral spur (Asterisk) and the TDW anterior to the scleral spur. (**B**) 6 mm Ologen sponge. (**C**) Suprachoroidal implantation of Ologen through the choroidal window. (**D**) Positioning of the Ologen as a placeholder in the suprachoroidal space, keeping the drainage open. (**E**) UBM imaging (50 MHz) of the implanted Ologen; note the suprachoroidal fluid around the implant. (**F**) Schematic showing the mechanism of ScD fluid circulation (arrows) and suprachoroidal Ologen (asterisk). Also marked are the scleral spur (star) and the ciliary body (square).

**Figure 7 jcm-12-03031-f007:**
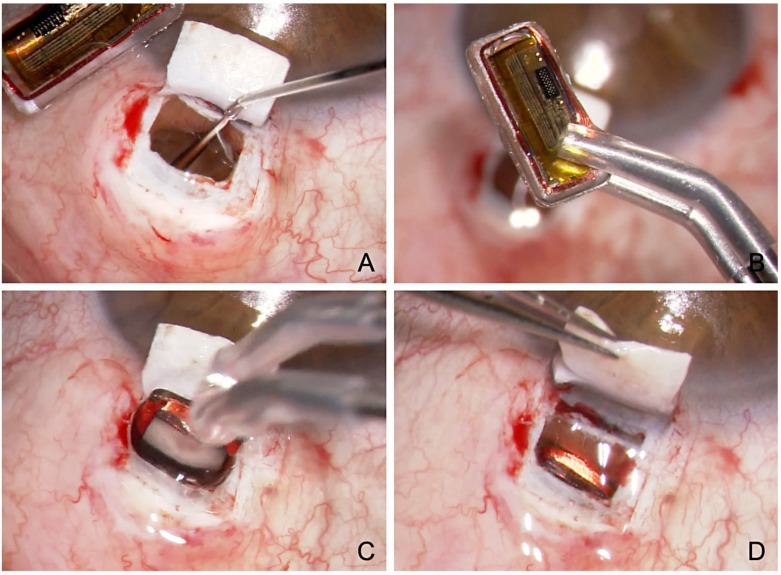
Surgical technique of suprachoroidal implantation of the Eyemate-SC sensor. (**A**) Viscoelastic is injected at the end of the CP with ScD to expand the suprachoroidal space (**B**) The Eyemate-SC sensor is grasped carefully with padded implantation forceps. (**C**) The sensor is implanted into the suprachoroidal space. (**D**) The Eyemate-SC sensor serves as a placeholder for suprachoroidal drainage while allowing continuous IOP monitoring.

**Figure 8 jcm-12-03031-f008:**
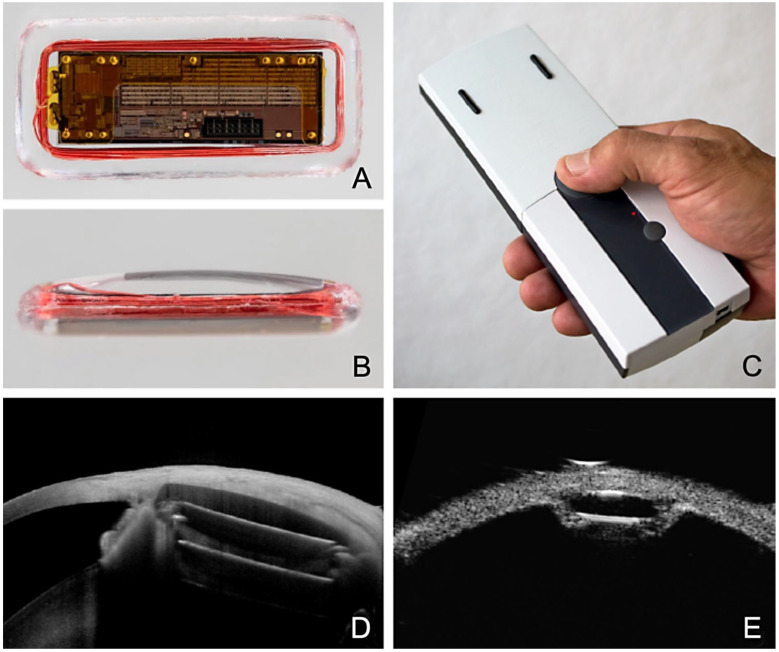
Eyemate-SC sensor looking at the electronics from (**A**) the top and (**B**) the side. (**C**) Mesograph readout device for wireless telemetric IOP measurement. Multimodal imaging 6 months after suprachoroidal implantation of the Eyemate-SC sensor. (**D**) An anterior segment (AC)-OCT visualizes the microelectronics carrier substrate, but not the silicone encapsulation. (**E**) The UBM image (50 MHz) depicts the lens-shaped rounded silicone encapsulation smoothly adapting to the curved scleral shape.

**Figure 9 jcm-12-03031-f009:**
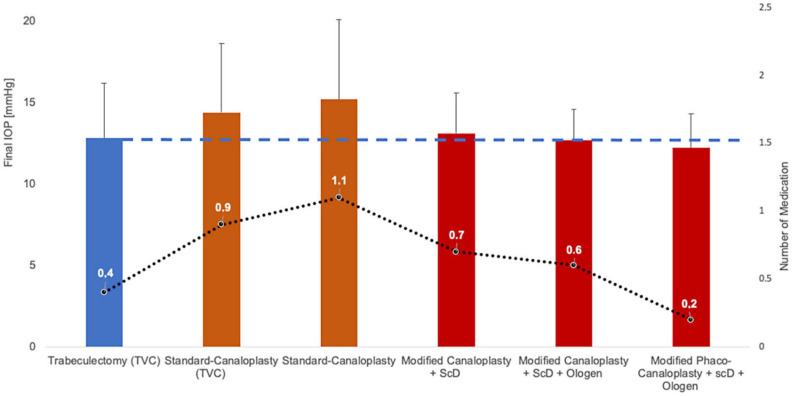
Evolution of canaloplasty modifications to improve the IOP-lowering success and the amount of remaining medication. This continuous refinement, especially the combination with ScD, allows CP to achieve IOP values comparable to TE (dashed line) and a low number of medications (dotted line) for the first time, but without the complications of TE [13,16,22,30,31,41].

**Table 1 jcm-12-03031-t001:** Rates of hypotony for different surgical techniques.

Long-Term Hypotony Rate (<5 mmHg for More than 90 Days)	
Canaloplasty	<1% [24]
PreserFlo	<1% [25]
Tubes	2.8% [26]
Trabeculectomy	18.8% [23]

## Data Availability

Not applicable.

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
