# Peer review of "Advances in Canaloplasty—Modified Techniques Yield Strong Pressure Reduction with Low Risk Profile"

_jcm, 2023, doi:10.3390/jcm12083031_

Round 1
Reviewer 1 Report
In this interesting review of the literature, the Authors thoroughly explore and discuss the modified techniques used to enhance the surgical outcomes of canaloplasty. The issue presented is however already widely discussed, as several reviews have already been published in the recent literature.
The manuscript appears to be clear in every passage and generally well written and structured. English language is appropriate and fully meets the Journal standards. References reported are appropriate as well, although the body text is missing references in several passages. Images are clear and illustrate in detail the steps of the surgical procedures presented. Image captions are well organized and properly describe the figures.
Below my recommendations:
Page 2: “Thus, the rationale of CP is to activate and improve natural outflow pathways (tra- becular facility)”. Add ref.
Page 2: “This effect dilates the canal to almost triple its original size, making it easily visible in the ul- trasound biomicroscope (UBM) 50 MHz”. Add ref.
Page 2: “The placement of one or two 10-0 Prolene tensioning sutures results in permanent stretching and tightening of the trabecular meshwork (i.e., surgical "pilocarpine effect"). Add ref.
Page 2: “Applying tension to the TDW results in the controlled percolation of aqueous hu- mor into the intrascleral cleft formed after resection of the deep flap, which can be en- hanced by YAG goniopuncture”. Add ref.
Page 3: “The detailed surgical technique has been described before”. Add ref.
Page 3. “Care is taken to avoid microfistulation under the conjunctiva or even the formation of a subconjunctival bleb”. Add ref.
Page 4. “Last”. Correct to “Lasts”.
Page 4. “An important new indication is preoperated eyes after failed TE”. Add ref.
Page 6: “Comparative studies with TE confirmed that both methods achieved a sufficient IOP- lowering effect and significant reduction of postoperative glaucoma medications”. Add ref.
Page 6. “[…] whereas CP patients required slightly more medication postoperatively but rarely had relevant complications”. Add ref.
Page 6. “[…] it is questionable whether such an ex- treme IOP reduction of up to 55% is really necessary or useful to these pre-damaged eyes”. Add ref.
Page 6. “In addition, there were antime- tabolite-associated complications such as corneal erosions and avascular filtering blebs”. Add ref.
Page 6. References [32] and [33] should be moved at the end of the sentence.
Page 7. “lower risk profile, easier follow-up, and lower impact on patient quality of life”. Add ref.
Page 7. “The goal of numerous scientific efforts has been to modify CP in a manner such that it achieves IOP values comparable to those obtained after TE”. Add ref.
Page 7. “[…] to achieve an additive IOP-lowering effect by improving uveoscleral facility”. Add ref.
Page 7. “With this Sulzbach modification”. What does it mean? Please clarify the passage.
Page 8. “[…] located directly anterior to the scleral spur can be prepared more reliably”. Add ref.
Page 8. “The uveoscleral drainage effect is additive to the trabecular effect of conventional CP”. Add ref.
Page 8. “These results were confirmed in a comparative two-armed study of 417 eyes over 12 months”. Add ref.
Page 9. “[…] the IOP was even slightly lower (12.9 ± 1.9 mmHg), suggesting a sus- tained effect”. Add ref.
Page 9. “Interestingly, modified CP with ScD and suprachoroidal Ologen also seems to be ef- fective in treating pseudoexfoliation glaucoma”. Add ref.
Page 9. Delete the graft next to the reference [16].
Page 10. “[…] to ensure continuous drainage of aqueous humor under the conjunctiva”. Add ref.
Page 10. “[…] but has all the disadvantages of a filtering surgery and so must be managed accordingly”. Add ref.
Page 10. “[…] postoperative hypotony (15%) was significantly higher than in the control group (1.1%). Add ref.
Page 10. Move ref [46] at the end of the sentence.
Page 10. “Thus, CP with adjuvant use of MMC seems to be a way to improve the efficacy of standard CP”. Add ref.
Page 10. “In the future, an important modification is the combination of CP with a supracho- roidal IOP sensor for telemetric self-measurement by the patient”. Add ref.
Page 10. “Particularly after glau- coma surgery, frequent and reliable measurement of the IOP is crucial to verify successful IOP adjustment in the target range”. Add ref.
Page 11. “Suprachoroidal implantation of the Eyemate-SC IOP sensor (7.5 × 3.5 mm and an out- wardly decreasing thickness of 1.3 mm in the center and 0.9 mm on the periphery) can be excellently combined with CP”. Add ref.
Page 11. “[…] but also allows continuous, postoperative IOP-monitoring simultaneously”. Add ref.
Page 11. “[…] that require strict control of the mean IOP and diurnal IOP fluctuations”. Add ref.
Page 11. “[…] (GAT) is limited owing to the altered corneal biomechanics. Add ref.
Page 12. “[…] and, if necessary, the therapy can be adapted accordingly”. Add ref.
Page 12. Move ref [50] at the end of the sentence.
Reviewer 2 Report
The authors have done a very well-designed review. The only thing which can add on the value of the manuscript would be more discussion and comparison with MIGS which are now are very popular procedures worldwide.
Reviewer 3 Report
The author draws attention to the fact that canaloplasty can be performed intelligently in different variations and combined with other modifications.
For combinations, there is always the problem that it becomes more difficult to assess the proportion of the individual components.
- Overall, it would probably be better if the author had limited himself to fewer. For example, the special situation of a bleb (filtration) in particular shows that too little consideration was given to the mutual influence of the different drainage paths. When aqueous humour drains outwards, the influence of the other changes (trans-trabecular and suprachoroidal) decreases.
- Is the use of the pressure sensor really part of the method?
- It is highly likely that suprachoroidal surgery also carries certain risks that should be mentioned (bleeding, hypotony).
- A fair comparison should be made. There is still not the same evidence with a number of RCTs and corresponding follow-up, similar to TE. Therefore, it is not yet possible to make a final judgement on the surgeon factor.
- For canaloplasty, the additional costs also need to be discussed.
In section 7, apart from the publication, there is talk of a two-arm "study". This should be stated more clearly. Have the data been submitted as a publication? Was it a prospective study or a retrospective evaluation of the experimental phase?
The photographs are very nicely prepared. Would it be possible to show the schematic drawing (Fig 6) in a more differentiated way (scleral spur, SC, etc.) and in different views?
